# Sleeve Gastrectomy Improves Hepatic Glucose Metabolism by Downregulating FBXO2 and Activating the PI3K-AKT Pathway

**DOI:** 10.3390/ijms24065544

**Published:** 2023-03-14

**Authors:** Ningyuan Chen, Ruican Cao, Zhao Zhang, Sai Zhou, Sanyuan Hu

**Affiliations:** 1Cheeloo College of Medicine, Shandong University, Jinan 250012, China; 2Graduate Faculty, Shandong First Medical University, Jinan 250117, China

**Keywords:** SG, T2DM, glucose metabolism, FBXO2, PI3K-AKT pathway

## Abstract

Type 2 diabetes mellitus (T2DM), a chronic metabolic disease, is a public health concern that seriously endangers human health. Sleeve gastrectomy (SG) can relieve T2DM by improving glucose homeostasis and enhancing insulin sensitivity. However, its specific underlying mechanism remains elusive. SG and sham surgery were performed on mice fed a high-fat diet (HFD) for 16 weeks. Lipid metabolism was evaluated via histology and serum lipid analysis. Glucose metabolism was evaluated using the oral glucose tolerance test (OGTT) and insulin tolerance test (ITT). Compared with the sham group, the SG group displayed a reduction in liver lipid accumulation and glucose intolerance, and western blot analysis revealed that the AMPK and PI3K-AKT pathways were activated. Furthermore, transcription and translation levels of FBXO2 were reduced after SG. After liver-specific overexpression of FBXO2, the improvement in glucose metabolism observed following SG was blunted; however, the remission of fatty liver was not influenced by the over expression of FBXO2. Our study explores the mechanism of SG in relieving T2DM, indicating that FBXO2 is a noninvasive therapeutic target that warrants further investigation.

## 1. Introduction

Type 2 diabetes mellitus (T2DM), a chronic metabolic disease characterized by hyperglycemia, impaired islet cell function, and insulin resistance, accounts for more than 90% of diabetic patients [1,2]. Obesity is a key risk factor for the occurrence and development of pre-diabetes and T2DM. In recent years, the incidence of obesity and T2DM in the world has rapidly increased [3]. Approximately 592 million people will suffer from diabetes worldwide by 2035 [4]. Poor treatment of T2DM leads to serious diseases involving multiple organ systems. Thus, obesity and T2DM have become public health concerns that seriously endanger contemporary human health [5].

Obese patients respond poorly to therapy and have a high failure rate following T2DM treatment due to changing lifestyles and drug administration [6]. The effectiveness and feasibility of bariatric surgery in treating T2DM are gradually being recognized by scholars at home and abroad; this procedure can lead to sustained weight loss and a high T2DM remission rate. The International Diabetes Federation (IDF) and the American Diabetes Association (ADA) named this procedure as one treatment method for diabetes [7,8]. Sleeve gastrectomy (SG) has become the most common bariatric surgery globally. Many studies have also shown that SG is not only a restrictive bariatric surgery but can also significantly relieve T2DM and other obesity-related complications, including improving glucose homeostasis and enhancing insulin sensitivity [9]. SG has been suggested to be superior to traditional drug therapy in terms of controlling blood glucose and improving or even reversing T2DM [10]. Following SG, the mechanism of metabolic improvement depends not only on significant weight loss but also on molecular mechanisms independent of weight loss [11]. The mechanisms of metabolic improvement caused by bariatric surgery focus on three main areas: improved secretion of gastrointestinal hormones such as GLP-1, improved bile acid metabolism, and changes in the gut microbiota [12,13]. These mechanisms can increase insulin sensitivity in the liver; however, the exact underlying mechanism remains unclear.

F-box protein is a key protein component of the SKP1-Cullin1-F-box protein (SCF) E3 ligase complex, which participates in the ubiquitination pathway and various biological processes [14]. F-box-only protein 2 (FBXO2) is a member of this family, which can regulate the ubiquitination pathway. Studies have shown that abnormal expression of FBXO2 degrades insulin receptors (IR) in obese mice through ubiquitination, thus disrupting glucose homeostasis [15]. In addition, it has also been reported that FBXO2 plays an important role in Parkinson’s disease and tumor development [16,17]. However, there is little research on FBXO2 and metabolic diseases, and its role in metabolic improvement after bariatric surgery is unclear. This study aimed to determine the changes in FBXO2 after sleeve gastrectomy and its role in glucose homeostasis improvement

## 2. Results

### 2.1. SG-Induced Weight Loss and Short-Term Reduction of Food Intake in Mice

After 16 weeks of consuming a high-fat diet, all C57 mice were randomly divided into SG and sham surgery groups. No obvious differences were found in body weight between groups before surgery. During the first 2 weeks post-surgery, the food intake of mice in the SG group was lower than that of the sham operation group; however, after 3 weeks, the food intake in the two groups showed no significant difference (Figure 1A).

The body weight curves showed that SG surgery induced obvious body weight loss in the first week after surgery in comparison to the sham surgery. Eight weeks after surgery, the SG group showed a greater reduction in weight loss and body size than the sham operation group (Figure 1B,C). We calculated body weight gain from the fourth to the eighth week in both groups and found that weight regain in the SG group was lower than that in the sham group despite identical food intake (Figure 1D).

### 2.2. SG Relieves HFD-Induced Body Glucose and Lipid Metabolism Disorder

We further studied lipid and glucose metabolism changes in mice after SG. H&E and ORO staining of liver sections showed that lipid accumulation was obviously reduced in the SG group (Figure 2A). Compared with the sham group, the serum triglyceride level of mice in the SG group decreased significantly. In addition, the serum total cholesterol level showed a slight decrease after SG, although the difference did not reach statistical significance (Figure 2B). We further examined the composition of apolipoproteins and found a significant increase in HDL-C and a distinct decrease in LDL-C after bariatric surgery (Figure 2C). This result was consistent with previous studies. Bariatric surgery could reduce cardiovascular risk events by altering apolipoprotein composition [18].

In glucose metabolism, the OGTT results showed that SG decreased fasting blood glucose and blood glucose levels after 15, 30, 60, and 120 min following 2 g/kg glucose gavage (Figure 2D). The ITT results showed that fasting blood glucose and blood glucose levels after 15, 30, 60, and 120 min following 1 U/kg insulin subcutaneous injection also decreased in SG group mice (Figure 2E). Further, the RT-qPCR results revealed that RNA levels of genes associated with lipogenesis (Fasn, Acly, and Pparg) and lipid uptake (CD36 and Fabp1) in mouse livers were downregulated in the SG group (Figure 2F). In terms of glucose metabolism, mRNA levels of insulin receptor substrate 2 (Irs2), Foxo1, as well as genes associated with glucogenesis (G6pc and Pck1) were downregulated in mouse livers of the SG group compared to those of the sham surgery group (Figure 2G).

We further studied the changes in signaling pathways that occurred alongside changes in lipid and glucose metabolism. Western blot analysis showed that the phosphorylation level of AMPKα was significantly increased in the SG group. Conversely, the phosphorylation level of mTOR, which plays a role in hepatic steatosis, was suppressed (Figure 3A). Our results also showed that following SG, the protein level of IR was upregulated and the PI3K-AKT-GSK3β pathway was activated, as evidenced by the increased phosphorylation levels of both AKT and GSK3β in SG groups (Figure 3B). Both of these promote insulin sensitivity and contribute to glucose homeostasis. Our data suggested that sleeve gastrectomy could significantly alleviate HFD-induced hepatic lipid accumulation and improve hepatic glucose metabolism.

### 2.3. SG Reduces Liver FBXO2 Level but Not by Reducing Blood Free Fatty Acid Concentration

FBXO2 belongs to the F-box family of proteins and can reportedly disrupt glucose homeostasis through direct interaction with the ubiquitin insulin receptor. It was upregulated in mice fed an HFD [15]. However, the expression of FBXO2 after clinical bariatric surgery in the livers of patients cannot be determined using this type of analysis. After SG in mice, the western blot and immunohistochemistry results showed that the protein levels of FBXO2 were downregulated in the liver (Figure 3C,D). In addition, qPCR results showed that the mRNA levels of FBXO2 were reduced (Figure 3E).

Further in vitro study on human HepG2 cells explored the mechanism of the observed downregulation of FBXO2 caused by SG with Oil red O and Nile red staining. When treated with free fatty acids (0.3 μmol sodium oleate), HepG2 cells showed obvious lipid accumulation (Appendix A). However, mRNA and protein levels of FBXO2 showed no significant change (Appendix A). SG can reduce blood levels of free fatty acids, and our study showed that the reduction in liver FBXO2 level was not affected by this phenomenon but was meditated by an alternative mechanism. Previous studies have shown that FBXO2 was regulated by the NFκB signaling pathway. GLP-1 is one of the main intestinal hormones that play a role in SG, and it has been reported that GLP-1 could reduce NFκB activation, which may cause changes in FBXO2 expression in the liver [12,19].

### 2.4. Hepatic-Specific Overexpression of FBXO2 Partly Reversed the Remission of Glucose Homeostasis Caused by SG

We overexpressed FBXO2 specifically in the liver by injecting AAV-His-FBXO2 into the tail vein following 16 weeks of HFD feeding and before SG, during the 8 weeks post-surgery. The food intake curve results showed that both the SG and SG FBXO2 overexpression groups showed a reduction in food intake at the first two weeks after surgery (Figure 4A). Both the SG and SG FBXO2 overexpression groups showed significant weight loss and reduction in body size compared to the sham operation group (Figure 4B,C). H&E and Oil red O staining of liver sections showed that lipid accumulation was obviously reduced in the SG and SG FBXO2 overexpression groups (Figure 4D). Compared with the sham group, the serum triglyceride levels of mice in the SG and SG FBXO2 overexpression groups decreased significantly. However, the serum T-Cho levels showed no significant difference among the three groups (Figure 4E). A significant increase in HDL-C and a distinct decrease in LDL-C was found in the SG and SG FBXO2 overexpression groups compared with the sham group (Figure 4E). In glucose metabolism, the OGTT and ITT results showed that fasting blood glucose and blood glucose levels at 15, 30, 60, and 120 min following gavage in the SG FBXO2 overexpression group were higher than those in the SG group, but they were lower than those in the sham group and showed statistical differences at some time points (Figure 4G,H). 

Compared with the SG group, the SG FBXO2 overexpression group showed no significant differences in food intake, body weight, and size (Figure 4A–C). Although overexpression of FBXO2 partly reversed the reduction in lipid accumulation caused by SG (Figure 4D), serum triglyceride, total cholesterol, HDL-C, and LDL-C showed no differences between the two groups (Figure 4E,F). However, overexpression of FBXO2 significantly increased blood glucose levels in OGTT and ITT (Figure 4F,G).

Mechanistically, overexpression of FBXO2 did not affect the phosphorylation levels of AMPKα and mTOR after SG (Figure 5A), but it could impair the IR and PI3K-AKT-GSK3β pathway (Figure 5B). Overexpression of FBXO2 significantly decreased the protein level of IR and reduced the protein level of PI3K and phosphorylation level of AKT, while GSK3β caused a decrease in PI3K-AKT signaling pathway activity. Furthermore, the RT-qPCR results showed that there were no significant differences in the mRNA levels of genes associated with lipid metabolism in the SG and SG FBXO2 overexpression groups (Figure 5C). Some of the genes associated with glucose metabolism (Foxo1, Irs2, and G6pc) in the SG FBXO2 overexpression group were upregulated compared to those in the SG group (Figure 5D). This result showed that FBXO2 overexpression could weaken the effect of SG on glucose metabolism through the PI3K-AKT-GSK3β pathway but did not affect the function of SG on lipid metabolism.

## 3. Discussion

Type 2 diabetes is a disease that develops rapidly [1]. It primarily manifests as an increase in fasting blood glucose and postprandial blood sugar, accompanied by insulin resistance in multiple organs [20]. A variety of therapeutic drugs have been used to treat type 2 diabetes, which mainly act by improving insulin resistance or directly supplementing insulin [21]. However, conventional drug treatment struggles to stably control and reverse the development of type 2 diabetes, and other organs may be damaged as a result of these therapeutics in the long term.

PI3K phosphatidylinositol 3 kinase was discovered in the 1990s [22] and it has been reported that PI3K/AKT signaling plays a very important role in cell physiology, mediating growth factor signaling in biological growth [23,24]. It plays a key role in cellular processes and regulates glucose homeostasis, lipid metabolism, protein synthesis, cell proliferation, and cell survival. AKT is regulated directly by PI3K and participates in the regulation of glucose and lipids [25]. In the intracellular chamber, AKT converts glucose into glucose-6-phosphate by stimulating hexokinase [26]. AKT regulates glycolysis by suppressing glucose-6-phosphate and glycogen synthase kinase 3 (GSK3) and reduces the expression of phosphoenolpyruvate carboxyl kinase (PEPCK) and glucose-6-phosphatase (G6PC) by inhibiting FoxO1, thus reducing gluconeogenesis and glucose levels [27,28,29,30]. In addition, AKT inhibits GSK3 through phosphorylation of GSKβ, thereby activating glycogen synthase (GS) and increasing glycogen synthesis [31]. This classical signaling pathway widely exists in many organs of the body, and it has been reported that the liver can reduce circulating glucose levels when receiving insulin through this mechanism [29]. In the fasting state, glucose is mainly used in the liver for gluconeogenesis and glycogen decomposition. It is then transported to various tissues while inhibiting the synthesis of fatty acids in these tissues. In the fed state, the PI3K/AKT signaling pathway reduces hepatic glucose production (HGP) and glycogen decomposition while increasing glycogen and fatty acid synthesis for storage and subsequent use by other tissues. Such a balance maintains the energy supply after meals and on an empty stomach. In obesity, the slow response of adipose tissue leads to a reduction in FFA uptake and glucose utilization, leading to ectopic accumulation in other tissues [32]. PI3K/AKT is impaired by ectopic lipid accumulation in the liver, insulin resistance, and nonalcoholic fatty liver disease (NAFLD), resulting in increased insulin resistance. Previous studies have shown that SG can effectively alleviate insulin resistance in type 2 diabetes [10,11]. In our experiment, the expression of PI3K in the liver increased after SG. In addition, AKT phosphorylation levels also increased significantly, which was consistent with previous studies [22,24]. However, the mechanism of how SG activated this pathway remained unclear in previous research.

IRS is one of the signaling proteins upstream of PI3K. When insulin binds to insulin receptor (IR), the inhibition of IRS is removed. It has been reported that mice with Irs2 gene knockout exhibited selective insulin resistance, while mice lacking Irs1 or Irs1 and Irs2 developed total insulin resistance [33,34]. This indicates that IRS1/2 is the cause of selective insulin resistance; in some cases, the insulin signal transduction in the liver remains intact, but the involuntary cellular pathway may be blocked, as described above, resulting in an increase in HGP during insulin resistance. One possible explanation is that the destruction of substrate IRS1/2 will cause insulin resistance. In addition, many studies have shown that the level of IRS1/2 increases after bariatric surgery [35,36]. These studies provide important insight into the mechanism of PI3K-AKT pathway activation after weight-loss surgery.

A previous study showed that FBXO2 is a member of the E3 ubiquitin ligase family and can target ubiquitin insulin signal receptor (IR) to disrupt the insulin signal pathway. Liver-specific overexpression and deletion of FBXO2 have been shown to affect glucose homeostasis [15]. The ubiquitin proteasome pathway is an important regulatory mechanism in vivo. In another study in our laboratory, we found that levels of FBXO2 protein and mRNA in the livers of obese patients were significantly upregulated. These results highlight the significant role of FBXO2 in the occurrence of insulin resistance and abnormal glucose metabolism. In the current study, we found that mRNA and protein levels of FBXO2 in mouse livers decreased significantly after bariatric surgery. The FBXO2-IR axis can be reversed to protect the PI3K-AKT signal pathway using this mechanism, which complements the mechanism that improves insulin sensitivity following SG surgery (Figure 6).

Type 2 diabetes can be treated with bariatric surgery, which is being performed with increasing regularity worldwide. There are many ways to perform bariatric surgery. At present, SG is still the most widely performed weight loss surgery in the world [37]. Current studies have shown that SG can play a role in regulating glucose metabolism and homeostasis through a variety of mechanisms [12,13,38]. SG can significantly affect the secretion of intestinal hormones, including GLP-1, GLP-2, PYY, and other hormones [39]. Of these, GLP-1 is the most extensively studied, and GLP-1 receptor agonists have been widely used in the clinical treatment of obesity and type 2 diabetes [40]. GLP-1 also plays an important role in SG by inhibiting appetite [41]. Our study showed that mice the SG group no longer displayed reduced food intake after 3 weeks. This may be related to the special eating habits of rodents. Previous studies have shown that reducing energy intake can significantly improve human metabolic function. However, with the same food intake, SG mice showed lower body weight regain. A recent study showed that IL-27 was upregulated and promoted increased adipose tissue thermogenesis after bariatric surgery, revealing a specific mechanism of weight loss independent of dietary restriction [42].

NAFLD and T2MD often coexist in clinical practice. The excessive accumulation of lipids is one of the most important factors affecting the insulin sensitivity of the liver. As a result, lipotoxicity will increase, resulting in local chronic inflammation and impaired insulin signaling [43]. SG surgery can significantly improve liver lipid accumulation and alleviate insulin resistance through various mechanisms. In the liver, the AMPKα-mTOR signaling pathway plays a regulatory role in glucose and lipid metabolism in the body [44]. Previous studies have shown that long-term mTOR activation will cause significant lipid accumulation, while AMPKα can inhibit the function of mTOR by enhancing the phosphorylation of raptor [45,46]. Our results showed that after weight-loss surgery, the phosphorylation levels of AMPKα were significantly increased and those of mTOR were inhibited. This may explain the reduction in liver lipid accumulation following SG. Our study found that overexpression of FBXO2 did not seem to affect AMPK activation. In addition, overexpression of FBXO2 did not seem to significantly affect the expression of genes related to liver lipid metabolism. However, the results of liver tissue staining showed that overexpression of FBXO2 caused an increase in liver lipid accumulation. We speculate that this phenomenon may be caused by FBXO2 indirectly through PI3K-AKT, or that FBXO2 has other mechanisms that do not depend on the above two pathways to regulate lipid metabolism. These findings warrant further investigation.

## 4. Materials and Methods

### 4.1. Animals and NAFLD Mice Models

Male C57BL/6 mice aged 7–8 weeks were acquired from SPF Biotechnology Co. (Beijing, China). All mice were housed at a suitable temperature under a 12-h light/dark cycle. All mice were fed a high-fat diet (HFD) containing 60% kcal from fat (D12492; Xiao Shu You Tai Biotechnology, Beijing, China) for 16 weeks to induce insulin resistance and fatty liver [47]. At the beginning of the experiment, 20 mice were fed HFD. Of these, 4 mice were excluded from the group because they weighed less than 35 g. The 16 mice were subsequently treated with SG operation (n = 8) and sham operation (n = 8). Two SG mice died within one week of the operation. After dissection, it was found that one mouse died of gastric fistula and the other death was probably due to pyloric obstruction. No mice in the sham operation group died. We finally selected 6 mice in the SG group and 6 randomly selected mice in the sham group. All mice were sacrificed after continued 10-week HFD feeding post-operation. All mice were euthanized after 12 h fasting to collect samples. The animal research was approved by the Ethics Committee for Animal Research at Shandong Provincial Qian Foshan Hospital (S328; 3 April 2020).

### 4.2. Surgical Procedures

The SG surgery has been described previously [48]. Mice were anesthetized with 3% isoflurane before the operation and maintained on 1.5% isoflurane anesthesia during the operation. During the surgery, 70% of the total stomach, including the entire gastric fundus, was removed in mice in the SG group (Appendix A). Mice in the sham group were identically anesthetized, and midline incision and stomach exposure were performed, similar to mice in the SG group, without removing any part of the stomach. The mice were resuscitated at 35 °C after surgery to prevent low-temperature damage. Post-operatively, saline was injected subcutaneously into the mouse’s back for rehydration. Mice were checked daily for survival for the first two weeks after surgery. Food intake and body weight were monitored weekly until the mice were euthanized for the collection of the samples.

### 4.3. Mouse Adeno-Associated Virus8 (AAV8) Construction and Injection

AAV-His-FBXO2 with TGB promoter was used to construct a liver-specific FBXO2 overexpression mouse model, with AAV-His-Vector serving as the control. At 8 weeks, the mice were administered a dose of 5  ×  10^12^ vector genome (μg)/kg by tail vein injection [49]. The AAVs used above were purchased from WZ Biosciences Inc. (Shandong, China). In this part of the experiment, after adeno-associated virus8 injection and HFD, we obtained 24 obese mice that met the body weight requirement (SG, n = 8; SG FBXO2 overexpression, n = 8; and sham operation, n = 8). Three mice died within two weeks of the operation (SG, n = 2; SG FBXO2 overexpression, n = 1). We finally selected 6 mice in the SG group, 6 randomly selected mice in the SG FBXO2 overexpression group, and 6 randomly selected mice in the sham group.

### 4.4. Blood Biochemical Analysis

Blood was collected from the tail vein of the mice. Mice serum was obtained after centrifugation of blood at 8000× *g* for 20 min at 4 °C. The serum levels of triglyceride (TG) total cholesterol (T-Cho), high-density lipoprotein cholesterol (HDL-C), and low-density lipoprotein cholesterol (LDL-C) were measured using standard determination kits according to the manufacturer’s instructions (Nanjing Jiancheng, Nanjing, China).

### 4.5. Oral Glucose Tolerance Test (OGTT) and Insulin Tolerance Test (ITT)

The OGTT and ITT were performed on mice in the SG and sham groups 9 and 10 weeks after the operation. For OGTT, 2 g/kg of glucose was administered orally, and for ITT, 1 U/kg of insulin was injected intravenously into the mice after a 6 h fast. Serum glucose levels were measured at 0, 30, 60, 90, and 120 min using a standard glucometer (Johnson & Johnson, New Brunswick, NJ, USA).

### 4.6. Histological Analysis

The liver tissue sections were stained with hematoxylin and eosin (H&E) (Solarbio, Beijing, China) and hepatic fat accumulation was determined using Oil Red O (ORO) staining (Sigma-Aldrich, St. Louis, MO, USA). A light microscope (IX-73; Olympus Corporation, Tokyo, Japan) was used to capture the histological images of the tissue sections.

### 4.7. Cell Lines

The human liver HepG2 cell line was used in this study and was cultured in Dulbecco’s Modified Eagle Medium (DMEM) with 10% fetal bovine serum (FBS) and 1% penicillin-streptomycin in a humidified chamber with 5% CO_2_ at 37 °C. The HepG2 cells were exposed to odium oleate (OA) at a final concentration of 0.3 mM for 24 h to establish an in vitro cell lipid deposition model. The hepG2 cell line was purchased from Shanghai Zhongqiao Xinzhou Biotechnology Co., Ltd. (Shanghai, China).

### 4.8. Western Blot Analysis

Briefly, liver tissues and HepG2 cells were lysed in RIPA lysis buffer containing protease and phosphatase inhibitors, followed by western blotting using standard procedures [50]. The primary antibodies used are listed in Appendix A. The immunoblotting results were quantified and statistically analyzed using ImageJ (National Institutes of Health, Bethesda, MD, USA) and GraphPad Prism software (GraphPad Software 8.0, San Diego, CA, USA), respectively.

### 4.9. RNA Preparation and Quantitative Real-Time PCR (qPCR)

Total RNA was extracted from cells using TRIzol reagent according to the manufacturer’s instructions and then reverse transcribed to generate cDNA (#RR047A; Takara Bio Inc., Shiga, Japan). The SYBR Green qRT-PCR method was applied to quantify real-time PCR amplification (LightCycler 480 II, Roche Diagnostics, Indianapolis, IN, USA). The expression levels were calculated using the Δct-method; some of the results were calculated using a log2 negative logarithm. The primer pairs used in our study are listed in Appendix A.

### 4.10. Oil Red O (ORO) and Nile Red Straining in HepG2 Cells

Oil red storage solution was prepared with 5 g/L Oil Red O and isopropyl alcohol as a solvent storage fluid. The working solution of ORO (ORO:deionized water = 2:3) was used for Oil Red O staining. Nile red stain was dissolved in acetone and diluted 1:10,000 in 1× PBS for Nile red straining. DAPI was used to stain cell nuclei.

### 4.11. Immunohistochemistry

Immunohistochemistry analysis was carried out on paraffin-embedded mouse liver tissue sections. A standard xylene ethanol procedure was used to dewax and rehydrate the tissue. The slices were boiled in citric acid buffer (pH 6.0) for 20 min. The activity of endogenous peroxidase was inhibited in methanol using 3% hydrogen peroxide (UltraVision Hydrogen Peroxide Block; Thermo Fisher Scientific, Waltham, MA, USA) for 15 min. Thereafter, the sections were incubated with FBXO2 antibody (Santa Cruz Biotechnology, Dallas, TX, USA) for 4 h at 37 °C. The HRP polymer (Ultra Vision Quanto Detection System; Thermo Fisher Scientific, Waltham, MA, USA) and DAB chromogen (DAB Peroxidase Substrate Kit; Vector Laboratories, Burlingame, CA, USA) were used to visualize FBXO2 immunoreactivity in the sections. Finally, the sections were counterstained with hematoxylin (ScyTek Laboratories, Logan, UT, USA) and dehydrated.

### 4.12. Statistical Analysis

The experimental results are expressed as mean ± standard deviation (SD) and were analyzed using GraphPad Prism version 8.3 software (GraphPad Software). Statistical differences between two experimental groups were evaluated using an unpaired Student’s *t*-test. Statistical differences among three experimental groups were evaluated using one- or two-way ANOVA. Statistical significance was considered at *p* < 0.05.

## 5. Conclusions

This study emphasized the key role of FBXO2 in metabolism. However, we did not investigate the direct factors affecting FBXO2 expression caused by SG. Previous studies have shown that FBXO2 may be affected by the NFκB-IKKβ pathway in the liver [13], and it has been reported that GLP-1 can inhibit the activation of NFκB [41]. We speculate that GLP-1 may be the cause of the downregulation of FBXO2 and could establish a new link in the interaction between liver inflammation and insulin resistance.

In conclusion, our experiments expand the current knowledge on epigenetic changes due to SG. We identified FBXO2 as an E3 ubiquitin ligase protein that plays a role in T2MD remission following SG. Our results supplement studies on the mechanisms of action of SG and provide new insight into the treatment of T2MD.

## Figures and Tables

**Figure 1 ijms-24-05544-f001:**
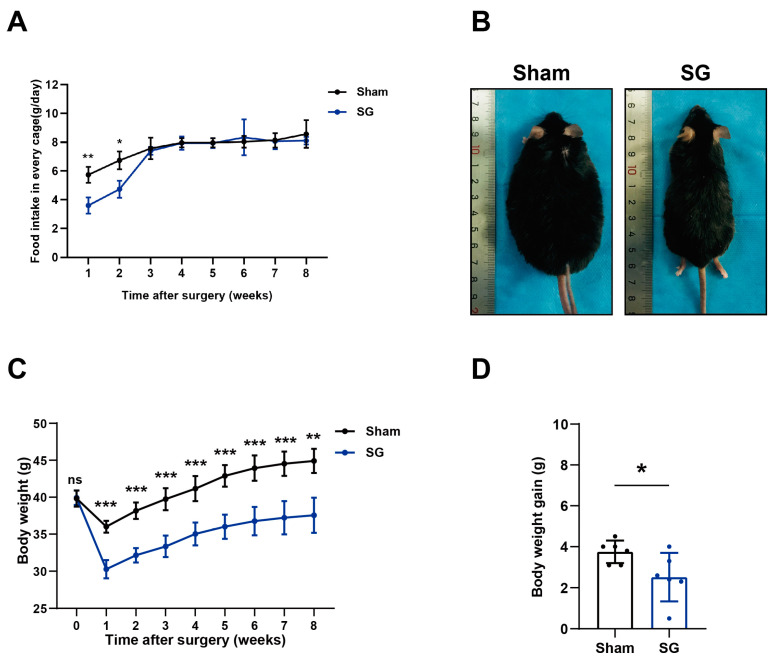
Sleeve gastrectomy(SG) alleviated obesity in mice. (**A**) Food intake curves after surgery. Food intake was calculated by weighing the change in HFD diet weight within one day, two mice were put in one cage (n = 3); (**B**) Representative image of mice at 8 weeks after surgery in each group. Images were taken of mice under anesthesia conditions; (**C**) Body weight curves of mice in SG group and sham surgery group after surgery (n = 6); (**D**) The amount of weight regained from 4 to 8 weeks after surgery (*t*-test result. * *p* < 0.05, ** *p* < 0.01, *** *p* < 0.001).

**Figure 2 ijms-24-05544-f002:**
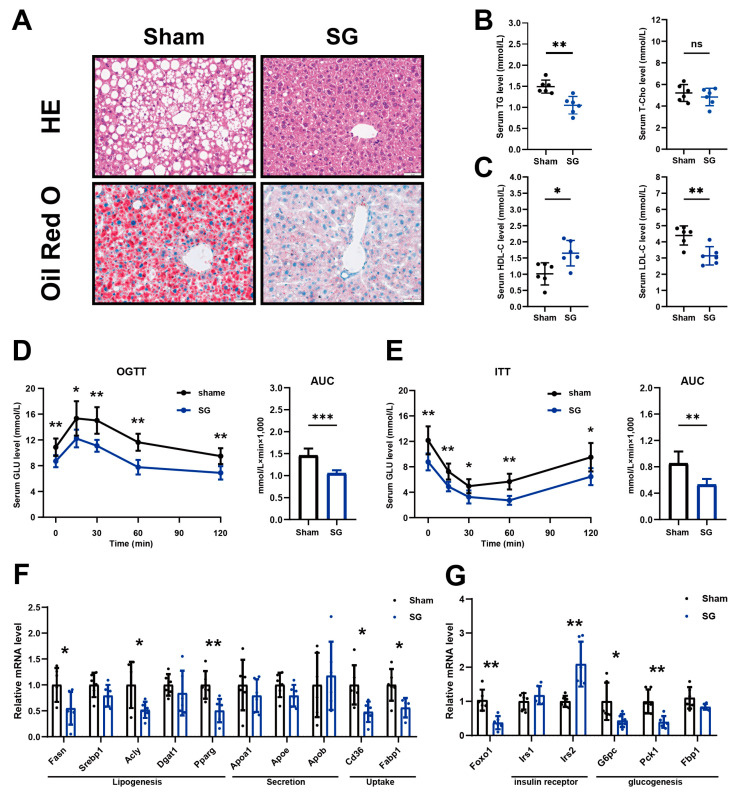
SG reduced hepatic steatosis and promoted glucose metabolism. (**A**) Representative images of H&E and Oil Red O staining of liver tissues of mice in sham and SG groups (scale bar, 50 μm); (**B**) Serum TG and T-Cho levels in each group. (n = 6); (**C**) Serum HDL and LDL levels in each group. (n = 6). Blood glucose level from glucose tolerance test (**D**) and insulin tolerance test (**E**). Areas under the curve (AUC) of blood glucose levels (n = 6); (**F**) Relative mRNA levels of genes involved in lipid metabolism, including lipogenesis genes, lipid uptake, and lipid secretion (n = 6); (**G**) Relative mRNA levels of genes involved in glucose metabolism, including insulin receptor and glucogenesis (n = 6) (*t*-test result. * *p* < 0.05, ** *p* < 0.01, *** *p* < 0.001).

**Figure 3 ijms-24-05544-f003:**
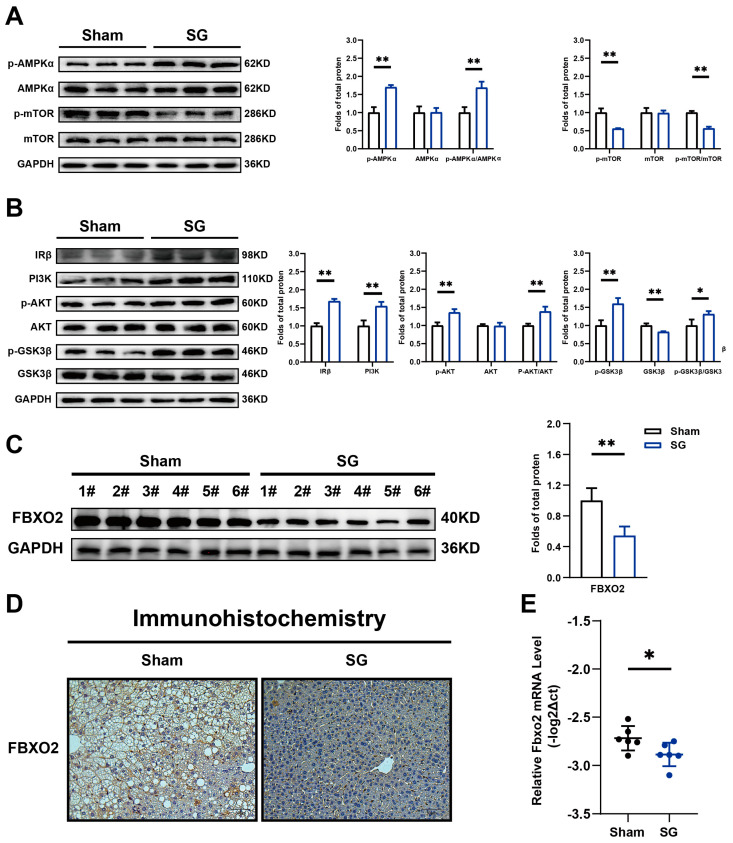
SG reduced FBXO2 level and improved signaling pathway related to lipid and glucose metabolism. Western blot result of (**A**) AMPK-mTOR pathway; and (**B**) IR and PI3K−AKT pathway in mouse livers of SG and sham groups; (**C**) Western blot result FBXO2 in mouse livers of SG and sham groups. (**D**) Representative images of immunohistochemical staining of FBXO2 in the liver tissues of SG and sham groups (scale bar, 50 μm); (**E**) Relative mRNA levels of Fbxo2 in mouse livers of SG and sham groups(n = 6) (*t*-test result. * *p* < 0.05, ** *p* < 0.01).

**Figure 4 ijms-24-05544-f004:**
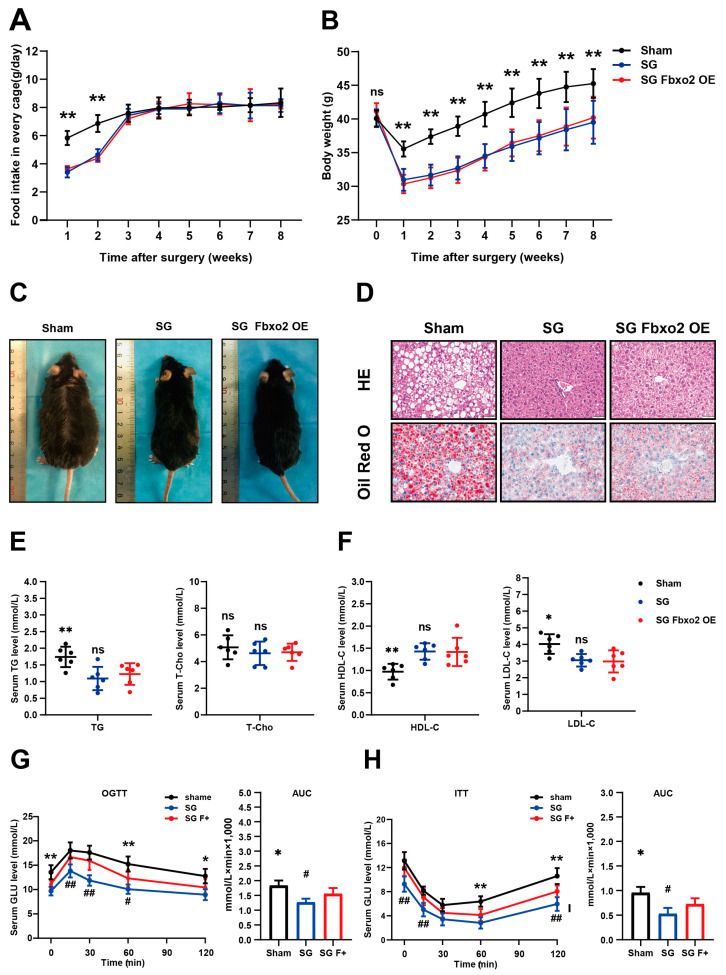
Liver-specific overexpression of FBXO2 blunted the improvement of glucose metabolism caused by SG. AAV-His-FBXO2 was used to crate FBXO2 liver-specific overexpression mice model; AAV-His-Vector served as control. The sham operation group served as a blank control. (**A**) Food intake curves after surgery. Food intake was calculated by weighing the change in HFD feed weight within one day; two mice were put in a cage (n = 3); (**B**) Body weight curves of mice in indicated group after surgery (n = 6); (**C**) Representative image of mice at 8 weeks after surgery in each group. All images were taken under anesthesia conditions; (**D**) Representative images of H&E and Oil Red O staining of liver tissues of mice in each group (scale bar, 50 μm); (**E**) Serum TG and T-Cho levels in each group. (n = 6); (**F**) Serum HDL and LDL levels in each group. (n = 6). Blood glucose levels of glucose tolerance test (**G**) and insulin tolerance test (**H**). Area under the curve (AUC) of blood glucose levels (n = 6) (ANOVA test results: * Sham group verses SG FBXO2 overexpression group, * *p* < 0.05, ** *p* < 0.01. # SG group verses SG FBXO2 overexpression group, # *p* < 0.05, ## *p* < 0.01; ns, no significance).

**Figure 5 ijms-24-05544-f005:**
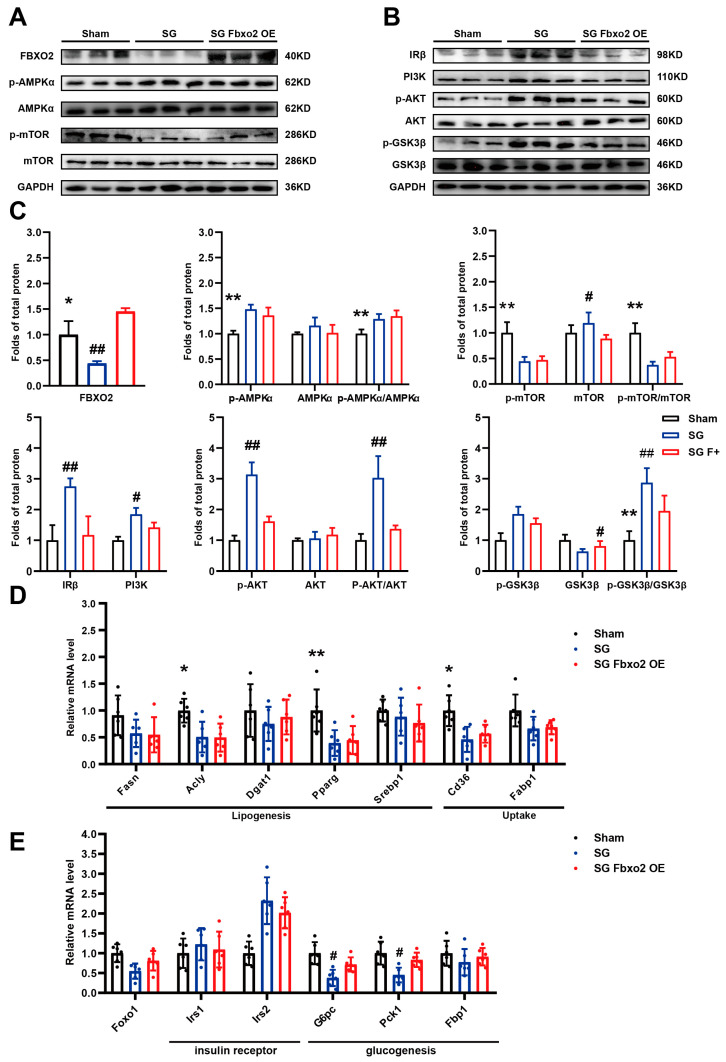
Liver-specific overexpression of FBXO2 reversed activation of the PI3K-AKT pathway induced by SG. AAV-His-FBXO2 was used to create the liver-specific FBXO2 overexpression mouse model, AAV-His-Vector served as control. Western blot results of (**A**) AMPK-mTOR pathway and (**B**) IR and PI3K-AKT pathway in mouse livers of indicated groups; (**C**) Analysis of western blot results (**A**,**B**) (n = 3); (**D**) Relative mRNA levels of genes involved in lipid metabolism, including lipogenesis genes, lipid uptake, and lipid secretion (n = 6); (**E**) Relative mRNA levels of genes involved in glucose metabolism, including insulin receptor and glucogenesis (n = 6) (ANOVA test results: * Sham group verses SG FBXO2 overexpression group, * *p* < 0.05, ** *p* < 0.01. # SG group versus SG FBXO2 overexpression group, # *p* < 0.05, ## *p* < 0.01).

**Figure 6 ijms-24-05544-f006:**
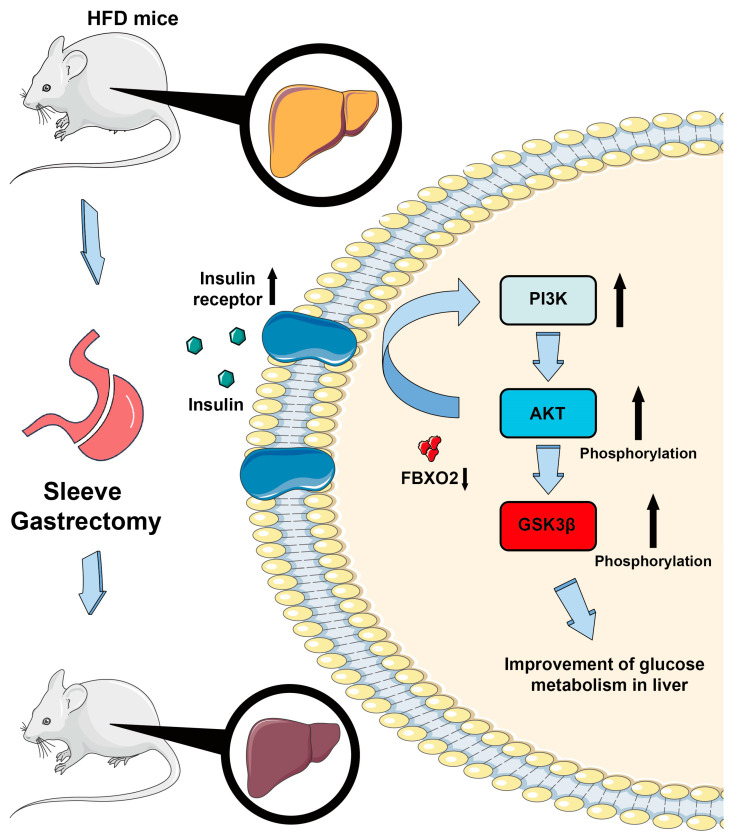
Working model of how sleeve gastrectomy reduces FBXO2 expression and activates the PI3K-AKT pathway. Sleeve gastrectomy (SG) reverses liver glucose and lipid metabolic disorders caused by HFD feeding, resulting in reduced blood glucose level and remission of fatty liver. Mechanically, SG causes downregulation of FBXO2, which protects insulin receptors from degradation, resulting in the PI3K-AKT pathway activation and glucose metabolism improvement.

## Data Availability

Data pertaining to specific variants generated during the downstream analyses, which support the findings of this study. are available upon request to the corresponding author C.H.). De-identified data will be made available upon request.

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
