# Peer review of "Sleeve Gastrectomy Improves Hepatic Glucose Metabolism by Downregulating FBXO2 and Activating the PI3K-AKT Pathway"

_ijms, 2023, doi:10.3390/ijms24065544_

Round 1

Reviewer 1 Report

The manuscript “ Sleeve gastrectomy improves glucose metabolism by downregulating FBOX2 and activating the PI3K-AKT pathway” by Chen et al reports a study that aims to delineate the mechanism to explain the metabolic improvement caused by bariatric surgery. By measuring the gene expression, lipid and glucose metabolism, hepatic signaling transduction, and in vivo overexpression in mice after gastrectomy, the authors concluded that the change of the FBOX2-IRS pathway is the major mechanism for the beneficial effect. Given that our understanding of the role of FBOX2 in metabolism is limited, the findings in this report might provide new insight into this area. 

In general, the manuscript is well-written with a logical presentation. Most data are convincing and support the major conclusion. Nevertheless, the author should address the following issue before it is suitable for publication.

  1. As the author stated, the FBOX2 modulated the insulin signaling via ubiquitinating the insulin receptor, it is surprising that the current study did not measure the amount of IR in the SG group. As an important piece of information that links SG and insulin sensitivity improvement, it is highly recommended the IR content should be measured. 
  2. The results are not described in order. For instance, Fig 2F was mentioned after Fig 2C. Similarly, Fig 3D was mentioned before Fig 3C. 
  3. Why hepatic lipid accumulation was reduced in the SC group, but their food intake was comparable with the control animals? Are there any changes in energy expenditure? The author should at least discuss this. 
  4. The author should explain why the serum TG level is significantly lower in the SG mice, but the serum cholesterol is comparable. Change in the composition of lipoproteins? 
  5. In Fig 3C, the amount of hepatic FBXO2 was significantly reduced after SG, but the level of FBOX2 in Fig 5A was comparable with the control. The author should explain this discrepancy of data. 
  6. A discussion on how SG reduces hepatic FBXO2 should be included in the text. Similarly, how the reduced FBOX2 content links with the change of PI3K/IRS2 expression  
  7. The legend of Fig 1A does not provide any information. Indeed, Fig 1A should not be included in the main text as it does not enhance our understanding of the scientific findings but should be presented as supplemental information.
  8. The legend of Fig 1D is incorrect.
  9. The legend of Fig 1E is missing.
  10. A reference should be included to support the change of GLP-1 in SG and its linkage with NFkB signaling. 
  11. Fig 4C shows the mice's outlook but not the H&E staining, as mentioned in the text (p.6). 

Author Response

We greatly appreciate the positive comments on our work (Sleeve gastrectomy improves glucose metabolism by downregulating FBXO2 and activating the PI3K-AKT pathway ijms-2235705). and the careful and detailed suggestions. We have now made extensive changes to our manuscript in response to the reviewers' suggestions, and these changes have greatly improved our overall research and manuscript.

Point-by-point response please see the attachment

Reviewer 2 Report

Generally

The authors studied the effect of Sleeve gastrectomy in improving glucose homeostasis and enhancing insulin sensitivity by studying e changes in FBXO2 in mice models of DIO. The work is interesting and well-written.

Methodology

·        The number of mice in each group should be mentioned in section 4.1

·        The reference number and date of the Ethics Committee approval should be mentioned.

·        Postoperative care and deaths should be reported.

·        When the mice were prepared for sample collection after surgery

·        I think that the methods section should be after the introduction based on the MDPI template

Results

·        Well written

Discussion

·        Well written.

Author Response

We are honored to receive your reviews of our study (Sleeve gastrectomy improves glucose metabolism by downregulating FBXO2 and activating the PI3K-AKT pathway ijms-2235705). We appreciate you for your positive comments on our research. We have read your comments on our study in detail and have made extensive revisions to the questions you have asked. These changes have been very helpful to our study.

Point-by-point response please see the attachment.

Reviewer 3 Report

In the manuscript entitled “Sleeve gastrectomy improves glucose metabolism by downregulating FBXO2 and activation the PI3K-AKT pathway”, the author investigated the effect of sleeve gastrectomy (SG) on glucose metabolism and found that AMPK and PI3K-AKT pathways are activated and AAV mediated overexpression of FBXO2 in the liver could blunt the improvement of glucose metabolism caused by SG. Although the finding is interesting, three are several issues that need to be addressed for further evaluation. My comments are as follows:

1)      The title of this manuscript should be modified since only liver tissue was investigated in the study.

2)      More background information on the mechanism of metabolic improvement caused by SG is needed in the “Introduction” section.

3)      The figure legend in Figure 2 is inconsistent with the information shown in Figure.

4)      The information in lines 99 to 102 in “Result” is inconsistent with the information presented in the “Material and Methods”, such as the glucose concentration, insulin concentration, and injection methods used for GTT and ITT.

5)      The Supplemental Figures were not provided for the evaluation.

6)      The information in lines 155 to 156 in “Result” is inconsistent with the information presented in Figure 4.

7)      The information in lines 161 to 162 in “Result” is inconsistent with the information in Figure 4C.

8)      The figure legend in Figure 4 is inconsistent with the information shown in the figure, including Figures 4B, 4C,4D, 4E, and 4F.

9)      Since FBXO2 is an E3 ligase complex-related protein, the protein levels of insulin receptors, not mRNA, should be detected in Figure 2G and Figure 5E.

10)  The working model presented in Figure 6 is overstated and more evidence is needed to support to get such a conclusion, especially about the ubiquitination of IRS by FBXO2 in this manuscript.

11)  English editing service is needed to correct grammar and typographical errors.

Author Response

We greatly appreciate the positive comments on our work (Sleeve gastrectomy improves glucose metabolism by downregulating FBXO2 and activating the PI3K-AKT pathway ijms-2235705). and the careful and detailed suggestions. We have now made extensive changes to our manuscript in response to the reviewers' suggestions, and these changes have greatly improved our overall research and manuscript.

Point-by-point response please see the attachment.

Round 2

Reviewer 1 Report

The authors have addressed all my concerns and the manuscript is now suitable for publication. 

Reviewer 3 Report

The author has addressed all my concerns and I have no further comments